# Characterization of Five New *Monosporascus* Species: Adaptation to Environmental Factors, Pathogenicity to Cucurbits and Sensitivity to Fungicides

**DOI:** 10.3390/jof6030169

**Published:** 2020-09-10

**Authors:** Allinny Luzia Alves Cavalcante, Andréia Mitsa Paiva Negreiros, Moisés Bento Tavares, Érica dos Santos Barreto, Josep Armengol, Rui Sales Júnior

**Affiliations:** 1Departamento de Ciências Agronômicas e Florestais, Universidade Federal Rural do Semi-Árido, Mossoró RN 59625-900, Brazil; cavalcanteallinny@gmail.com (A.L.A.C.); deia_mitsa@hotmail.com (A.M.P.N.); sestavaresagro@gmail.com (M.B.T.); ericasb13@hotmail.com (É.d.S.B.); 2Instituto Agroforestal Mediterráneo, Universitat Politècnica de València, Camino de Vera S/N, 46022 Valencia, Spain; jarmengo@eaf.upv.es

**Keywords:** fungicides, mycelial growth, pathogenicity, pH, salinity, soilborne fungi, virulence

## Abstract

In this study, five new recently described *Monosporascus* species, *M*. *brasiliensis*, *M*. *caatinguensis*, *M*. *mossoroensis*, *M*. *nordestinus*, and *M*. *semiaridus,* which were found on weeds collected from cucurbit cultivation fields in northeastern Brazil, are characterized regarding mycelial growth at different pH levels and salinity (NaCl) concentrations, their pathogenicity to selected cucurbit species, and their sensitivity to fungicides with different modes of action. Our results reveal great variability among the representative isolates of each *Monosporascus* spp. All of them showed a wide range of tolerance to different pH levels, and NaCl significantly reduced their in vitro mycelial growth, although no concentration was able to inhibit them completely. In pathogenicity tests, all seedlings of cucurbits evaluated, melon, watermelon, cucumber, and pumpkin, were susceptible to the five *Monosporascus* spp. in greenhouse experiments using artificial inoculation of roots. Moreover, all *Monosporascus* spp. were highly susceptible to the fungicides fludioxonil and fluazinam. Our findings provide relevant information about the response of these new *Monosporascus* spp. to environmental factors, plant genotypes and fungicides.

## 1. Introduction

Technological advances employed in cucurbits cultivation such as the use of hybrid cultivars, transplanted seedlings, plastic mulch, drip irrigation, and increased plant density have allowed the intensification of melon (*Cucumis melo* L.) and watermelon crops (*Citrullus lanatus* [Thunb.] Matsum. and Nakai). However, this has been also directly related to an increasing incidence of root diseases, such as *Monosporascus* root rot and vine decline (MRRVD), which is one of the major factors currently limiting the production and expansion of melon and watermelon crops worldwide [1,2,3]. The main symptoms of MRRVD on cucurbits manifest close to harvest, when yellowing, wilting and drying of the leaves occur, followed by a sudden vine decline, which causes the death of the plants and important economic loss [3].

MRRVD is caused by two soilborne ascomycetes belonging to the genus *Monosporascus*: *M*. *cannonballus* Pollack and Uecker and *M*. *eutypoides* (Petrak) von Arx, which colonize the roots of cucurbits, causing extensive necrotic and rotted areas, in which black perithecia, corresponding to the sexual reproductive structures of these pathogens, can be observed [3,4]. These fungal species were suggested to be conspecific, but it was demonstrated that they are distinct, with *M*. *cannonballus* being the most widespread [4].

*Monosporascus cannonballus* is a cosmopolitan pathogen which, to date, has been associated with MRRVD of cucurbits in 22 countries worldwide [3,5,6,7,8,9]. However, *M*. *eutypoides* has been unequivocally reported only in Tunisia from roots of watermelon and cucumber crops (*Cucumis sativus* L.) [4]. Recently, Negreiros et al. [10] conducted a survey in northeastern Brazil to investigate the role of prevalent weeds present in cucurbit fields as alternative hosts for fungal root pathogens. These authors found five new *Monosporascus* species on roots of *Boerhavia diffusa* L. and *Trianthema portulacastrum* L., which were described based on morphology and multilocus DNA sequence analyses: *M*. *brasiliensis* A. Negreiros, M. León, J. Armengol and R. Sales Júnior, *M*. *caatinguensis* A. Negreiros, M. León, J. Armengol and R. Sales Júnior, *M*. *mossoroensis* A. Negreiros, M. León, J. Armengol and R. Sales Júnior, *M*. *nordestinus* A. Negreiros, M. León, J. Armengol and R. Sales Júnior, and *M*. *semiaridus* A. Negreiros, M. León, J. Armengol and R. Sales Júnior. *Monosporascus* has been also reported as a common root endophyte in surveys of grasses, shrubs and herbaceous plants located in the southwestern United States using both molecular and culturing methods [11,12,13,14].

*Monosporascus cannonballus* is the most well-known species of the genus. There is some knowledge about the adaptability of *M*. *cannonballus* populations to different environmental factors such as temperature, hydrogen potential (pH) and salinity [1,3,15,16,17,18,19,20,21,22,23,24,25,26]. In fungi, these factors profoundly affect their growth rate but also can act as triggers in development pathways [27]. This information has been inferred from in vitro studies conducted with collections of *M. cannonballus* isolates obtained in areas where this fungus is a prevalent cucurbit pathogen.

*Monosporascus cannonballus* is considered a thermophilic fungus, well adapted to arid and semi-arid conditions, with optimal growth temperature for mycelial growth ranging from 25 to 35 °C; some isolates have been reported to grow at temperatures above 40 °C, and to be inhibited at temperatures below 15 °C [1,3]. Yasuaki et al. [19] showed that *M*. *cannnonballus* can tolerate an immersion in hot water at 50 °C for up to three days, but it died after 90 min immersion in hot water at 60 °C. Its optimal pH for mycelial growth ranges between 6 and 7; however, it was confirmed that this fungus can also tolerate a pH of 9, but its mycelial growth is highly reduced at pH 5, and it is totally inhibited at a pH below 4 [16]. Regarding salinity, *M*. *cannonballus* shows a high tolerance. In vitro tests showed that mycelium of this fungus can tolerate relatively high concentrations of NaCl and CaCl_2_ ranging from 8−10% of these compounds [1]. Recent studies conducted by Rhouma et al. [26] corroborated these previous studies, demonstrating that soils with high salt content may be favorable for MRRVD disease.

*Monosporascus cannonballus* is a melon and watermelon pathogen, but its pathogenicity to other Cucurbit species has also been explored. Cucumber, summer squash (*Cucurbita pepo* L.), pumpkin (*C*. *moschata* [Duch.] Duch. ex Poir), winter squash (*C*. *maxima* Duch.), bottle gourd (*Lagenaria siceraria* [Molina] Standl.), and loofah (*Luffa aegyptiaca* Mill.) have already been described as being susceptible to this pathogen in host range studies with artificial inoculation [15,28,29].

Control of *M. cannonballus* is difficult and the management programs against this pathogen integrate different management strategies, including the use of fungicides [30,31,32,33]. The fungicides fluazinam and kresoxim methyl completely inhibited the vegetative growth of *M*. *cannonballus* in vitro and fluazinam was able to suppress MRRVD in the field, with results ranging from 87 to 32% in disease control [34]. Guimarães et al. [35] conducted greenhouse studies with fluazinam, concluding that this active ingredient (a.i.) can be recommended in small doses (1.0 L/ha) to control MRRVD in melon crops. Doses of 10 µg/L a.i. of fluazinam were also effective in inhibiting the in vitro mycelial growth of 57 Brazilian isolates of *M*. *cannonballus* [25]. Azoxystrobin, prochloraz, and pyraclostrobin + boscalid exhibited high and similar efficacies in the control of *M. cannonballus* in field experiments, but fludioxonil was less effective than azoxystrobin and, in certain cases, showed some phytotoxicity to melon plants [31].

Much less information is available for *M*. *eutypoides*. This fungus has a mean optimum growth temperature of 29.43 ± 0.03 °C and only its pathogenicity to melon, watermelon and cucumber has been confirmed [4].

Regarding the five new *Monosporascus* species recently described in Brazil, *M*. *brasiliensis*, *M*. *caatinguensis*, *M*. *mossoroensis*, *M*. *nordestinus*, and *M*. *semiaridus*, what it is known so far is that their optimal growth temperatures range from 30 to 33 °C, and none of them was able to grow in potato-dextrose-agar (PDA) at temperatures of 10 and 40 °C [10]. Thus, the objective of this study was to obtain new phenotypic and pathogenicity information for these *Monosporascus* species by evaluating: (i) their mycelial growth at different pH levels and salinity concentrations; (ii) their pathogenicity to different cucurbits; and (iii) their sensitivity to several fungicides with different modes of action. In all these experiments, *M*. *cannonballus* was included as the reference species for the genus *Monosporascus*.

## 2. Materials and Methods

### 2.1. Monosporascus spp. Isolates

Six *Monosporascus* spp. isolates were used in this study (Table 1). Their identity was previously confirmed by molecular techniques [10], and all of them were deposited in the Collection of Phytopathogenic Fungi “Prof. Maria Menezes” (CMM) at the Universidade Federal Rural de Pernambuco (Recife, PE, Brazil). Prior to use, these isolates were grown in Petri dishes with PDA (Merck KGaA, Darmstadt, Germany) at 25 °C in darkness for 7–10 days.

### 2.2. Effect of pH on the Mycelial Growth Rate of Monosporascus spp.

The effect of pH on the mycelial growth rate of all isolates was determined using cultures grown on PDA. Mycelial plugs (8 mm in diameter) obtained from the growing edge of 10-day-old colonies were transferred to the center of PDA plates (one plug per plate) which were adjusted to pH 5, 6, 7, 8, and 9 with the addition of 1.5 N hydrochloric acid (HCl) or 1 N sodium hydroxide (NaOH). Plates were incubated in the dark at 28 °C. There were five replicates for each isolate and pH combination. The diameter of each colony was measured twice perpendicularly when it reached at least two thirds of the plate or at 7 days of growth, and the data were used to calculate the mycelial growth rate (MGR) as cm per day (cm/d). The experiment was conducted twice and was set up with a completely randomized design. One-way analysis of variance (ANOVA) was performed with data obtained from MGR, and the optimum pH (O_pH_) for MGR of each isolate was plotted against pH and a curve was fitted by a cubic polynomial regression (y = a + bx + cx^2^ + dx^3^) using TableCurve 2D v. 5.01 (Systat Software, Inc., San Jose, CA, USA).

### 2.3. Effect of Salinity on the Mycelial Growth of Monosporascus spp.

The effect of salinity on the mycelial growth of all isolates was determined using cultures grown on PDA. Mycelial plugs (8 mm in diameter) obtained from the growing edge of 10-day-old colonies were transferred to the center of PDA plates (one plug per plate) with the following concentrations of NaCl: 0, 250, 500, 750, and 1000 mM [36]. Plates were incubated in the dark at 28 °C. The experiment was conducted twice and was set up with a completely randomized design, with five replicates for each isolate. Mycelial growth rate was evaluated as described before and used to calculate the percentage of growth inhibition (PGI). The PGI of each isolate at different NaCl concentrations were subjected to a regression analysis using TableCurve 2D v. 5.01 (Systat Software, Inc., San Jose, CA, USA) and half-maximal effective concentration (EC_50_) was determined.

### 2.4. Pathogenicity of Monosporascus spp. to Cucurbits

The pathogenicity of *Monosporascus* spp. isolates to cucurbits was evaluated in a pot assay. Fungal inoculum was prepared following the method described by Ben Salem et al. [37], with some modifications. Wheat seeds were autoclaved in flasks three times at 120 °C for 1 h, with an interval of 24 h. Mycelial plugs (8 mm in diameter) of each isolate were used for inoculation of the seeds, which were incubated at 25 °C for four weeks until complete colonization by the pathogens. Flasks containing the inoculated seeds were agitated manually once a week to avoid inoculum clustering. The substrate used in the pots was composed of a mixture of sandy-clay soil passed through a 2 mm mesh and Tropstrato HT Hortaliças^®^ (Vida Verde, Brazil), at a proportion of 2:1. This mixture was autoclaved twice at 120 °C for 1 h, with an interval of 24 h. For soil infestation, approximately 12 g of the seeds colonized with each isolate were added in pots containing 2 kg of the sterile soil-substrate mixture. Only autoclaved noncolonized wheat grains were added to the substrate in the controls. One week after inoculation, 10-day-old seedlings of cucumber cv. ‘Racer’, melon cv. ‘Titannium’, pumpkin cv. ‘Mírian’ and watermelon cv. ‘Manchester’, were transplanted to pots containing infested soil. There were five pots per *Monosporascus* spp. and cucurbit species combination, with one seedling each. The pots were arranged in a complete randomized experimental design in a greenhouse at an average temperature of 35 °C, under natural daylight conditions, and watered thrice a week at field capacity. The experiment was conducted twice.

Disease evaluation was performed 50 days after the transplant. Plants were carefully removed from the pots, and the root systems were gently washed with tap water. Disease incidence in each cucurbit species was determined as the total number of infected plants from each *Monosporascus* spp. and expressed as a percentage. Disease severity was assessed using a diagrammatic scale adapted from Aegerter et al. [38], where: 0 = symptomless; 1 = less than 10% of the roots with weak discoloration or lesions; 2 = moderate discoloration or rot, with lesions reaching 25 to 35% of the roots; 3 = lesions converging to 50% of the roots and death of secondary roots; and 4 = generalized necrosis of the roots or dead plant. For isolation, roots were surface sterilized for 1 min in a 2.0% sodium hypochlorite solution and washed twice with sterile water. Seven root segments per plant from apparently affected areas were transferred to PDA supplemented with 500 mg/L of streptomycin sulphate.

The results of incidence and severity were analyzed with the non-parametric Kruskal–Wallis test at the probability level of 5% (*p* < 0.05) using the software Assistat, version 7.7 [39].

In addition, root and shoot lengths (RL and SL) of each plant, and the fresh and dry root (FRW and DRW) and shoot (FSW and DSW) weights were also measured. Dry weights were obtained by placing plant parts individually in paper bags, which were introduced in a forced circulation oven at 70 °C until a constant dry weight was reached. Data were submitted to ANOVA and means compared by Tukey at 5% probability using the Assistat software, version 7.7 [39].

### 2.5. Sensitivity of Monosporascus spp. to Fungicides

The effect of fungicides on mycelial growth of *Monosporascus* spp. was determined in vitro as described by Tonin et al. [40]. The treatments included five a.i.—boscalid (Cantus WG, 50% a.i., systemic, BASF S.A.), carbendazim (Carbendazim, 99.9% a.i., systemic, Syngenta Proteção de Cultivos Ltd.a, Sao Paulo, Brazil), cyprodinil (Unix 750 WG, 75% a.i., systemic, Syngenta Proteção de Cultivos Ltd.a), fluazinam (Frowncide 500 SC, 50% a.i., contact, ISK Biosciences do Brasil Defensivos Agrícolas Ltd.a), and fludioxonil (Maxim, 25% a.i., contact, Syngenta Proteção de Cultivos Ltd.a)—and five concentration levels: 0.01, 0.1, 1, 10 and 100 mg/L a.i. PDA plates without fungicide were used as controls. Mycelial plugs (8 mm in diameter) obtained from the growing edge of 10-days-old isolates of each *Monosporascus* spp. were transferred to the center of PDA plates containing the different fungicide concentration combinations and were incubated at 28 °C in darkness during 7 days. A completely randomized experimental design was used with five replicates per fungicide, concentration and *Monosporascus* spp. Colony diameters (cm) were measured in two perpendicular directions. The experiment was conducted twice. A preliminary ANOVA was performed to determine whether there were significant differences between the two repetitions of the experiment and whether the data could be combined. The TableCurve 2D v. 5.01 (Systat Software, Inc., San Jose, CA, USA) was used to determine the EC_50_ of PGI for each fungicide and *Monosporascus* spp. combination, using a four-, and three-parameter logistic model by plotting Probit transformed values of fungicide concentration and PGI, respectively.

## 3. Results

### 3.1. Effect of pH on Mycelial Growth of Monosporascus spp.

There was no significant effect of the experiment repetitions (ANOVA, *p* > 0.05), thus the data were combined. The cubic polynomial regression (y = a + bx + cx^2^ + dx^3^) selected to describe the mycelial growth at different pH levels, adjusted MGR data with R^2^ > 0.95 for all *Monosporascus* spp. isolates (Figure 1). All *Monosporascus* spp. grew at all pH levels. O_pH_ for mycelial growth varied between 5.72 (*M*. *caatinguensis*) and 8.05 (*M*. *cannonballus*). *Monosporascus brasiliensis*, *M*. *mossoroensis*, *M*. *nordestinus*, and *M*. *semiaridus* had optimal pH values of 7.53, 7.99, 6.52 and 6.85, respectively.

### 3.2. Effect of Salinity on Mycelial Growth of Monosporascus spp.

There was no significant effect of the experiment repetitions (ANOVA, *p* > 0.05), thus the data were combined. The adjusted means of NaCl concentrations, subjected to a regression analysis, showed significant positive correlations with R^2^ > 0.98 for all *Monosporascus* spp. isolates (Figure 2). All *Monosporascus* spp. grew in all NaCl concentrations, but this growth was reduced at the highest concentrations evaluated. The salinity concentrations that inhibit 50% of mycelial growth of *Monosporascus* spp. varied between 903.69 (*M*. *cannonballus*) and 994.41 mM (*M*. *mossoroensis*). The remaining species, *M. brasiliensis*, *M*. *caatinguensis*, *M*. *nordestinus*, and *M*. *semiaridus*, had EC_50_ values of 971.88, 961.02, 975.49 and 950.64 mM, respectively.

### 3.3. Pathogenicity of Monosporascus spp. to Cucurbits

There was no significant effect of the experiment repetitions (ANOVA, *p* > 0.05) for all variables, thus the data were combined. The inoculation of cucurbits with *Monosporascus* spp. caused significant statistical effect on disease incidence by *Monosporascus* spp. in cucumber (χ^2^ = 39.73; *p* < 0.05), pumpkin (χ^2^ = 46.64; *p* < 0.05), melon and watermelon seedlings (χ^2^ = 69.00; *p* < 0.05) (Table 2). In melon and watermelon, all the species inoculated caused a disease incidence of 100%, with all plants infected. In cucumber, the highest incidence (100%) was caused by *M*. *caatinguensis*, *M*. *cannonballus* and *M*. *semiaridus*, while in pumpkin, the highest incidence was caused by *M*. *caatinguensis*, *M*. *cannonballus*, *M*. *mossoroensis* and *M*. *nordestinus*.

Significant statistical effect was also observed for disease severity in cucumber (χ^2^ = 35.87; *p* < 0.05), melon (χ^2^ = 32.57; *p* < 0.05), pumpkin (χ^2^ = 42.23; *p* < 0.05) and watermelon seedlings (χ^2^ = 31.66; *p* < 0.05) (Table 2). In cucumber, the highest mean disease severity (3.30) was caused by *M*. *semiaridus*, while in melon, *M*. *brasiliensis*, *M*. *cannonballus*, *M*. *mossoroensis*, and *M*. *semiaridus* produced the highest mean disease severity (1.60). In pumpkin, the highest mean disease severity (1.80) was caused by *M*. *mossoroensis,* while in watermelon, *M*. *nordestinus* produced the highest mean disease severity (1.90). Thus, these were considered the most virulent *Monosporascus* spp. to each corresponding cucurbit species. The lowest mean disease severity was caused by *M*. *mossoroensis* in cucumber (0.40), *M*. *nordestinus* in melon (1.20), *M*. *brasiliensis* in pumpkin (0.60), and *M*. *semiaridus* in watermelon (1.30), thus these were considered the least virulent *Monosporascus* spp. to each corresponding cucurbit species.

In cucumber, the shorter RL (17.40 cm) was observed in the inoculation with *M*. *semiaridus*, which also caused the smallest FRW (7.94 g), SL (28.55 cm), FSW (11.94 g) and DSW (2.00 g) (Table 3). For DRW, all species differed statistically from the control. Melon presented shorter RL after the inoculation with *M*. *cannonballus* (19.44 cm), followed by *M*. *semiaridus* (19.90 cm) (Table 3). For FRW and DRW, all species differed statistically from the control, and shorter SL was observed the inoculation with *M*. *mossoroensis* (73.80 cm). The FRW and DSW did not differ statistically, with a coefficient of variation (CV) of 22.66 and 29.03%, respectively. In pumpkin, all species differed statistically from the control to RL and DRW, and the lowest FRW values were obtained in *M*. *caatinguensis*, *M*. *cannonballus* and *M*. *nordestinus* (8.69, 9.47, and 10.11 g, respectively) (Table 3). For SL, a lowest value was observed for *M*. *cannonballus* inoculation (13.62 cm), which was also observed for FSW (32.00 g). DSW showed the lowest values after the inoculation with *M*. *cannonballus*, *M*. *semiaridus* and *M*. *brasiliensis* (3.40, 3.49, and 3.63 g, respectively). Finally, the shorter RL value in watermelon was observed in the inoculation with *M*. *brasiliensis* (25.40 cm), which also showed lower FRW with *M*. *mossoroensis*, *M*. *caatinguensis* and *M*. *cannonballus* (4.39, 4.17, 4.01, and 3.94 g, respectively) (Table 3). All species differed statistically from the control to DRW and did not differ from each other to SL, FSW and DSW, with a CV of 31.14, 33.00 and 20.85%, respectively.

### 3.4. Sensitivity of Monosporascus spp. to Fungicides

There was no significant effect of the experiment repetitions (ANOVA, *p* > 0.05) for each fungicide, thus the data were combined. The effects of different fungicides on mycelial growth of *Monosporascus* spp. isolates are shown in Figure 3. Four-parameter logistic equations were adjusted and the EC_50_ values were calculated. The coefficients of determination ranged from 0.83 to 0.99. The mean EC_50_ for boscalid was 19.14 mg/L a.i. and the values of this variable ranged from 4.17 (*M. cannonballus*) to 41.69 mg/L a.i. (*M. caatinguensis*). For carbendazim, the mean EC_50_ was 4.58 mg/L a.i. and the values ranged from 0.17 (*M. brasiliensis*) to 8.32 mg/L a.i. (*M. nordestinus*). For the fungicide cyprodinil, the mean EC_50_ was 12.74 mg/L a.i. and the values ranged from 2.19 (*M. cannonballus*) to 23.44 mg/L a.i. (*M. brasiliensis*). For fluazinam, the mean EC_50_ was 0.34 mg/L a.i. and the values ranged from 0.04 (*M. cannonballus*) to 0.95 mg/L a.i. (*M. brasiliensis*). The mean EC_50_ for fludioxonil was 0.035 mg/L a.i., and the values of this variable ranged from 0.01 (*M. cannonballus*, *M. mossoroensis*, *M. nordestinus*, and *M. semiaridus*) to 0.09 mg/L a.i. (*M. brasiliensis*). Of the fungicides evaluated, those with lower EC_50_ for all *Monosporascus* spp. studied were fludioxonil and fluazinam (0.03 and 0.34 mg/L a.i., respectively).

## 4. Discussion

The main objective of this research was to obtain new biological information about five recently described *Monosporascus* species, regarding mycelial growth at different pH levels and salinity concentrations, their pathogenicity to cucurbits, and their sensitivity to fungicides with different modes of action.

Our results reveal great variability among the representative isolates of each species included in this study.

The optimal pH for mycelial growth of *Monosporascus* spp. ranged from 5.72 to 8.05, showing a wide range of tolerance, in agreement with previous studies conducted with *M*. *cannonballus* [1,23,24]. These values correspond with those indicated as a suitable soil pH range for cucurbits cultivation [41].

The presence of NaCl significantly reduced the in vitro mycelial growth of all *Monosporascus* spp. studied, and although no concentration was able to completely inhibit its growth, EC_50_ was above 900 mM for all species, indicating a moderately high tolerance to this substance. Previous studies have shown that *M*. *cannonballus* tolerates a high salinity content of NaCl or CaCl_2_ solutions when evaluated in vitro [1]. Reduction in mycelial growth in vitro by NaCl has also been observed in other cucurbit root fungal pathogens such as *Macrophomina phaseolina* (Tassi) Goid. and *M*. *pseudophaseolina* Crous, Sarr and Ndiaye [36,42,43]. The salinity stress is a major environmental constraint in semi-arid cucurbit-growing regions such as northeastern Brazil, where these new *Monosporascus* species were found. Exposure of the fungal cells to saline stress implies both exposure to specific osmotic stress that restricts water availability, and ion toxicity due to their ability to inhibit specific metabolic pathways [44].

MRRVD caused by *M. cannonballus* and *M. eutypoides* occurs mainly on melon, watermelon, and cucumber crops, although other cucurbit species have been shown to be susceptible in artificial inoculation studies [3,4,15,28,29]. In a similar way, in our study, the seedlings of all cucurbits evaluated were susceptible to *M*. *brasiliensis*, *M*. *caatinguensis*, *M*. *mossoroensis*, *M*. *nordestinus*, and *M*. *semiaridus*. It is interesting to note that to date, these new *Monosporascus* species have only been found as being associated to weed roots in cucurbit cultivation fields [10]. Although pot experiments cannot be used to predict the results in the field, our results suggest that more attention should be paid to these fungal species as potential cucurbit pathogens. Bruton [2] indicated that inadequate crop rotation contributes more than other factors to increase the inoculum of cucurbit soilborne pathogens, thus determining the emergence of new diseases or increasing disease incidence and severity of the already existing ones. Moreover, in the specific case of *Monosporascus* spp., Robinson et al. [14] commented that there is no evidence that isolates of *Monosporascus* from the roots of plants in natural ecosystems cause disease symptoms, despite their broad host association, thus raising questions regarding whether presumed endophytic lineages differ from pathogenic lineages with respect to specific genes or groups of genes. These authors compared the genomes of endophytic and pathogenic isolates within the genus *Monosporascus* and also across genera within the Xylariales with respect to genes for carbohydrate-active enzymes, genes known to be involved in pathogenicity in certain fungi, and genes for effector proteins that facilitate the colonization of plant tissues. Their results show that endophytic *Monosporascus* isolates from New Mexico contain more predicted genes associated with pathogenesis and host plant interactions than their agricultural relatives.

Our pathogenicity tests were conducted using an inoculum density (12 g of the seeds colonized/2 kg of the sterile soil-substrate mixture) lower than the one recommended by Ben Salem et al. [37] (200 g of inoculum/kg of peat). Andrade et al. [45] studied the influence of inoculum density of 44 isolates of *M*. *cannonballus* on the severity of MRRVD to melon. These authors concluded that low inoculum densities (0.1, 0.5, and 1.0 colony-forming unit (CFU)/g soil) produced high levels of disease, and this severity level did not increase when densities were increased. More recently, Castro et al. [46], evaluated the response of different melon genotypes to inoculation with *M*. *cannonballus* and *M*. *eutypoides* in greenhouse experiments in three different years, by using an inoculum dose of 200 g of inoculated wheat seeds/kg of substrate. They found a strong influence of temperature conditions in the different years of experiments on the incidence and severity of the disease caused by these pathogens on melon roots. This could have also influenced our results, because of the high average temperature in the greenhouse (35 °C) and the general thermophilic nature of *Monosporascus* spp. that could have favored root infection [4,10,17].

For *M*. *semiaridus* and *M*. *brasiliensis* in cucumber, the severity of the disease was highly correlated with a reduction in shoot and root length, and also for fresh and dry weights of roots and shoots, being therefore considered the most aggressive species for this cucurbit species. These variables were previously shown as being useful to evaluate the severity of the disease caused by *M*. *cannonballus* in cucurbits [37,46]. Differences in the root system, shoot length and dry weights were also observed in pumpkin, with emphasis on infection by *M*. *cannonballus*. Sales Júnior et al. [29] studied the reactions of cucurbits such as cucumber, melon, pumpkin and watermelon, after artificial inoculation with *M*. *cannonballus*. These authors observed that in all cucurbit species, there were root lesions, and it was also possible to observe perithecia of the pathogen. In our experiment, despite the severe damage caused to the root system in melons and watermelons, the dry weight of the plants was not affected, different from what was previously reported by several authors, and it was not possible to observe perithecia of the pathogens [47,48].

*Monosporascus* spp. sensitivity to fungicides was measured by EC_50_, which is specific and constant for a given a.i. and pathogen, and a low EC_50_ value represents a high fungicidal power [40,49]. All *Monosporascus* spp. were highly susceptible to fludioxonil and fluazinam fungicides, exhibiting EC_50_ values below 1 mg/L a.i. Fludioxonil and fluazinam also showed good in vitro efficacy against *M*. *cannonballus* in experiments performed by Pivonia et al. [31]. Contact fungicides such as fludioxonil and fluazinam do not have the ability to penetrate the tissues, acting as a barrier that protects the propagating structures from infection, being able to eradicate the pathogens found on the surface, and have been widely used in the management of root pathogens [50]. Boscalid, carbendazim and cyprodinil, with systemic modes of action, were moderately fungitoxic to *Monosporascus* spp, according to the criteria proposed by Edgington et al. [51] to frame substances with respect to their fungitoxicity. Boscalid and cyprodinil were moderately fungitoxic to all *Monosporascus* spp., and carbendazim, although moderately fungitoxic to all the species in general, showed high fungitoxicity to *M*. *brasiliensis* and *M*. *caatinguensis.*

To date, *M*. *brasiliensis*, *M*. *caatinguensis*, *M*. *mossoroensis*, *M*. *nordestinus*, and *M*. *semiaridus* have been only found in northeastern Brazil associated to weeds growing in cucurbit fields. But, managing soil-borne fungal diseases is a matter of understanding complex species interactions with the soil and host plant microbiome, and their response to environmental factors, plant genotypes and different control measures. The findings of this study provide relevant information about the behavior of these new *Monosporascus* spp.

## Figures and Tables

**Figure 1 jof-06-00169-f001:**
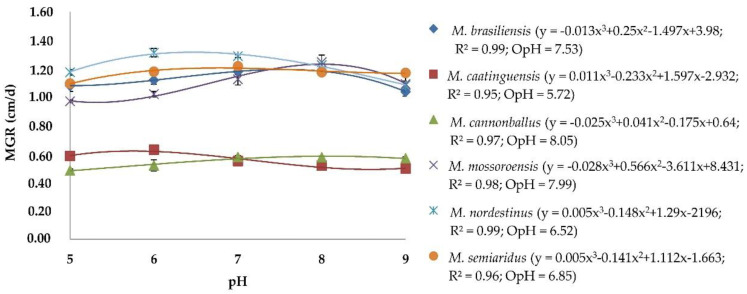
Regression equation, coefficient of determination (R^2^) and optimal pH for mycelial growth (O_pH_) of *Monosporascus* spp. isolates. y = adjusted with the values of the mycelial growth rate (MGR) at pHs 5, 6, 7, 8, and 9. O_pH_ = optimal hydrogen ion potential for mycelial growth of *Monosporascus* spp. calculated from the regression equation.

**Figure 2 jof-06-00169-f002:**
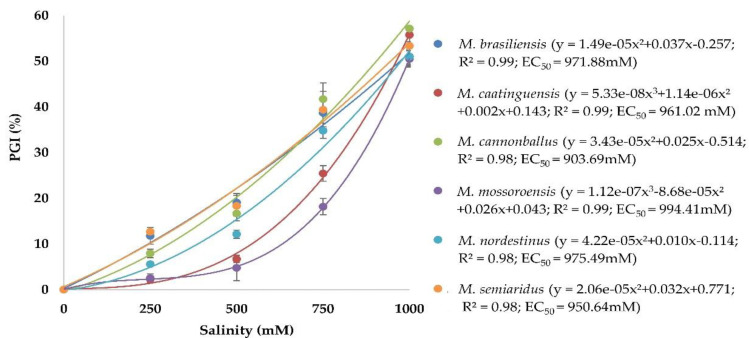
Regression equation, coefficient of determination (R^2^) and salinity half-maximal effect concentration (EC_50_) of Monosporascus spp. isolates. y = adjusted with the values of the percentage of growth inhibition (PGI) at NaCl concentrations of 0, 250, 500, 750, and 1000 mM. EC_50_ = salinity concentration that inhibits 50% of mycelial growth of *Monosporascus* spp. calculated from the regression equation (mM).

**Figure 3 jof-06-00169-f003:**
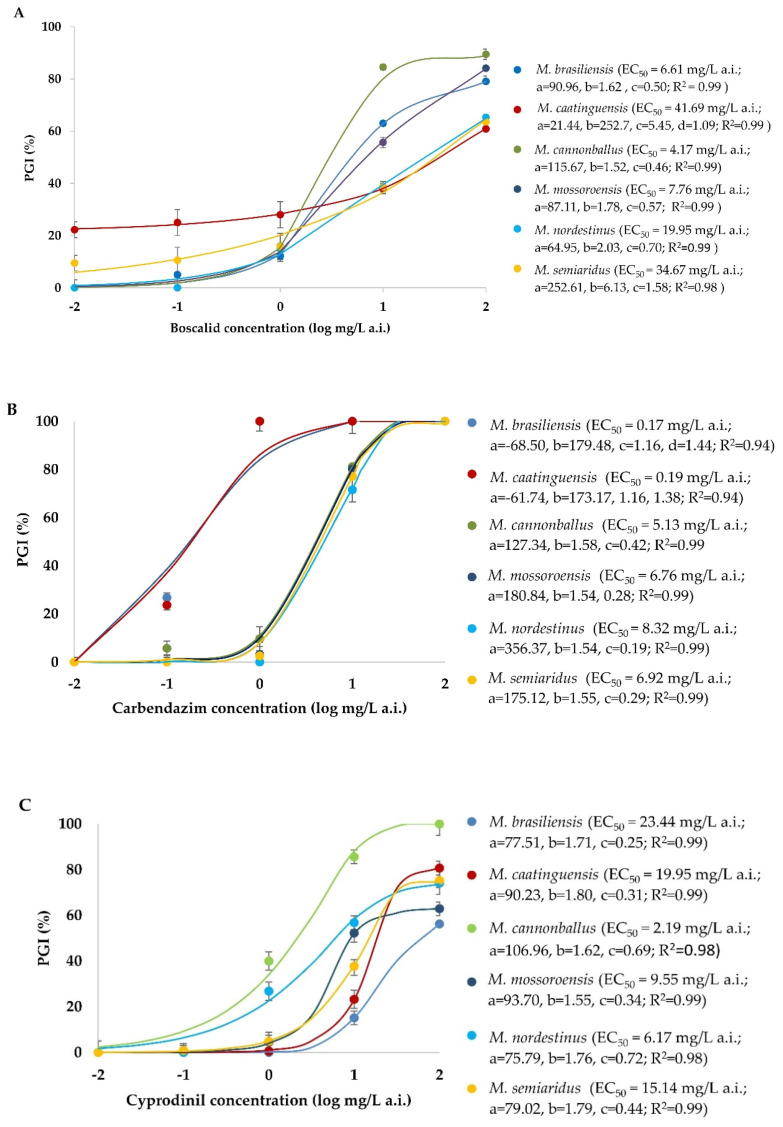
Regression equation, coefficient of determination (R^2^) and half-maximal effect concentration (EC_50_) of each *Monosporascus* spp. for the fungicides (**A**) boscalid, (**B**) carbendazim, (**C**) cyprodinil, (**D**) fluazinam, and (**E**) fludioxonil. y = adjusted with the values of percentage of growth inhibition (PGI) at concentrations of 0.01, 0.1, 1, 10 and 100 mg/L a.i. per fungicide. EC_50_ = 50% mycelial growth inhibition concentration calculated from the regression equation (mg/L).

**Table 1 jof-06-00169-t001:** Isolates of the *Monosporascus* spp. used in this study.

*Monosporascus* Species	Code (CMM) ^1^	Host	Location ^2^	GenBank ITS Region ^3^
*M. brasiliensis*	4839	*Trianthema portulacastrum*	Brazil, RN	MG 735234
*M. caatinguensis*	4833	*Boerhavia diffusa *	Brazil, CE	MG 735228
*M. cannonballus*	2429	*Cucumis melo*	Brazil, RN	JQ 762366
*M. mossoroensis*	4857	*Trianthema portulacastrum*	Brazil, RN	MG 735252
*M. nordestinus*	4846	*Trianthema portulacastrum*	Brazil, RN	MG 735241
*M. semiaridus*	4830	*Trianthema portulacastrum*	Brazil, CE	MG 735222

^1^ CMM = Culture Collection of Phytopathogenic Fungi “Prof. Maria Menezes” of the Universidade Federal Rural de Pernambuco (Recife, PE, Brazil). ^2^ CE = Ceará state and RN = Rio Grande do Norte state. ^3^ Sequence of the Internal Transcribed Spacer Region (ITS) of the isolates deposited at GenBank.

**Table 2 jof-06-00169-t002:** Incidence and severity of the disease in cucumber, melon, pumpkin, and watermelon seedlings by *Monosporascus* spp.

	Cucumber				Melon			
Treatments	Disease Incidence		Disease Severity		Disease Incidence		Disease Severity	
	Rank ^1^	Mean (%)	Rank ^1^	Mean (%)	Rank ^1^	Mean (%)	Rank ^1^	Mean (%)
*M. brasiliensis*	32.50 ab	60	39.35 bc	2.10	40.50 b	100	41.90 b	1.60
*M. caatinguensis*	46.50 b	100	37.75 abc	1.30	40.50 b	100	36.75 b	1.30
*M. cannonballus*	46.50 b	100	42.55 bc	1.60	40.50 b	100	43.45 b	1.60
*M. mossoroensis*	25.50 ab	40	20.90 ab	0.40	40.50 b	100	43.45 b	1.60
*M. nordestinus*	39.50 b	80	39.00 bc	1.60	40.50 b	100	34.00 b	1.20
*M. semiaridus*	46.50 b	100	57.45 c	3.30	40.50 b	100	43.45 b	1.60
Control	11.50 a	0	11.50 a	0.00	5.50 a	0	5.50 a	0.00
χ^2^	39.73		35.87		69.00		32.57	
	**Pumpkin**				**Watermelon**			
*M. brasiliensis*	29.50 b	60	25.60 ab	0.60	40.50 b	100	38.00 b	1.70
*M. caatinguensis*	43.50 b	100	47.65 bc	1.60	40.50 b	100	41.20 b	1.70
*M. cannonballus*	43.50 b	100	37.00 bc	1.00	40.50 b	100	42.80 b	1.80
*M. mossoroensis*	43.50 b	100	48.30 bc	1.80	40.50 b	100	44.80 b	1.80
*M. nordestinus*	43.50 b	100	50.15 c	1.70	40.50 b	100	44.40 b	1.90
*M. semiaridus*	36.50 b	80	31.30 abc	0.80	40.50 b	100	31.80 b	1.30
Control	8.50 a	0	8.50 a	0.00	5.50 a	0	5.50 a	0.00
χ^2^	46.64		42.23		69.00		31.66	

χ^2^ = significant chi-square values; values followed by the same letter in the columns do not present statistical difference between them by the non-parametric Kruskal–Wallis test (*p* < 0.05). ^1^ Average of the ranks for all observations within each sample. Data are mean values of two experiments, each with five replications (pots) per treatment and one plant per replication.

**Table 3 jof-06-00169-t003:** Effect of *Monosporascus* spp. inoculation on root and shoot length, fresh root and shoot weight, and dry root and shoot weight, of cucumber, melon, pumpkin and watermelon seedlings.

Treatments	Cucumber					
	RL ^1^ (cm)	FRW ^2^ (g)	DRW ^3^ (g)	SL ^4^ (cm)	FSW ^5^ (g)	DSW ^6^ (g)
*M. brasiliensis*	21.30 c	14.24 ab	0.37 b	46.90 b	27.47 a	2.95 bc
*M. caatinguensis*	35.40 a	12.65 ab	0.62 b	60.80 a	36.26 a	4.88 a
*M. cannonballus*	30.80 ab	9.90 bc	0.47 b	64.40 a	31.33 a	4.37 ab
*M. mossoroensis*	32.08 ab	12.27 abc	0.56 b	62.90 a	33.79 a	4.03 ab
*M. nordestinus*	24.60 bc	10.50 bc	0.44 b	68.00 a	29.84 a	3.91 ab
*M. semiaridus*	17.40 c	7.94 c	0.45 b	28.55 c	11.94 b	2.00 c
Control	39.80 a	17.52 a	1.00 a	67.12 a	37.19 a	4.90 a
CV (%)	23.78	27.04	37.67	16.37	28.72	30.39
	**Melon**					
*M. brasiliensis*	27.00 ab	3.84 b	0.28 b	97.20 a	46.51 a	6.06 a
*M. caatinguensis*	25.46 abc	7.08 b	0.36 b	97.22 a	46.73 a	6.86 a
*M. cannonballus*	19.44 c	4.32 b	0.35 b	89.82 ab	50.60 a	6.49 a
*M. mossoroensis*	24.70 abc	5.68 b	0.31 b	73.80 b	38.47 a	5.21 a
*M. nordestinus*	22.90 abc	6.43 b	0.40 b	106.75 a	51.72 a	7.67 a
*M. semiaridus*	19.90 bc	4.91 b	0.36 b	85.00 ab	47.31 a	5.99 a
Control	28.80 a	12.44 a	1.32 a	108.00 a	43.05 a	6.12 a
CV (%)	22.07	38.89	35.21	18.05	22.66	29.03
	**Pumpkin**					
*M. brasiliensis*	36.50 b	11.48 bc	0.67 b	16.10 ab	32.41 b	3.63 c
*M. caatinguensis*	37.80 b	8.69 c	0.66 b	18.75 a	38.24 ab	4.76 ab
*M. cannonballus*	35.73 b	9.47 c	0.67 b	13.62 b	32.00 b	3.40 c
*M. mossoroensis*	36.11 b	10.70 bc	0.69 b	16.76 ab	32.22 b	3.82 bc
*M. nordestinus*	31.78 b	10.11 c	0.66 b	17.18 ab	36.35 ab	4.23 abc
*M. semiaridus*	35.80 b	15.11 b	0.81 b	17.80 ab	34.40 b	3.49 c
Control	50.60 a	20.02 a	2.57 a	18.70 a	45.78 a	4.91 a
CV (%)	22.32	26.58	34.20	18.57	21.83	18.81
	**Watermelon**					
*M. brasiliensis*	25.40 b	4.39 c	0.30 b	96.90 a	32.72 a	4.55 a
*M. caatinguensis*	28.96 ab	4.01 c	0.27 b	100.05 a	31.04 a	3.94 a
*M. cannonballus*	26.50 ab	3.94 c	0.29 b	102.80 a	35.87 a	5.04 a
*M. mossoroensis*	27.50 ab	4.17 c	0.30 b	109.24 a	36.50 a	4.45 a
*M. nordestinus*	29.80 ab	7.26 b	0.36 b	120.20 a	38.58 a	4.84 a
*M. semiaridus*	28.85 ab	5.50 bc	0.38 b	103.70 a	37.48 a	5.20 a
Control	34.11 a	11.67 a	1.57 a	103.08 a	36.44 a	5.25 a
CV (%)	19.62	35.32	36.64	31.14	33.00	20.85

CV (%) = significant coefficient of variation values; values followed by the same letter in the columns do not present statistical difference between them by the Tukey test (*p* < 0.05). Data are mean values of two experiments, each with five replications (pots) per treatment and one plant per replication. ^1^ Root length. ^2^ Fresh root weight. ^3^ Dry root weight. ^4^ Shoot length. ^5^ Fresh shoot weight. ^6^ Dry shoot weight.

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
