# Peer review of "Characterization of Five New Monosporascus Species: Adaptation to Environmental Factors, Pathogenicity to Cucurbits and Sensitivity to Fungicides"

_jof, 2020, doi:10.3390/jof6030169_

Round 1
Reviewer 1 Report
Review report
Characterization of five new Monosporascus species: adaptation to environmental factors, pathogenicity to cucurbits and sensitivity to fungicides
The work includes a report on five recently described Monosporascus species, regarding mycelial growth at different pH levels and salinity concentrations, their pathogenicity to cucurbits sprouts (50 DAS), and their sensitivity to fungicides with different modes of action.
The manuscript consists of a lot of experimental work, and, from that point of view, it deserves merit.
The findings of this work are very interesting and important and contribute greatly to our knowledge about those Monosporascus species and their harmful potential as pathogens. The results advance our understanding of the pathogen isolates and their relationships with the host plant. The results are also essential to the development of disease management strategies.
However, I consider the manuscript has some weaknesses that should be carefully addressed.
General comments
My main concern is that it should be noted that the Monosporascus species ability to become cucurbit pathogens is also dependent on their ability to survive in the soil and compete with the soil and host plant microbiome, during a full growth period. Also, the ability of fungicides tested here to restrict those fungi species may be very different in field conditions. However, the findings presented here are a good opening stage for future studies. This issue should be carefully referred to in the text, and should be discussed in the Discussion.
The Introduction and the Methods are well written, but specific points should be addressed.
The presented data in the Result section should be rework and improve, as will specify below.
Results interpretation should be made carefully since the number of repeats is very limited.
The Discussion is well written, very interesting. Please note that salinity pressure is actually two tresses: osmotic pressure and ionic pressure. Please add a paragraph that discusses this.
Also, as stated above, it should be taken into consideration that the ability of pots experiments to predict the results in the field is limited. Please add a paragraph in the Discussion that discusses this.
Specific comments
Lines 23 – you stated that your findings “suggesting their potential to become cucurbit pathogens” and latte you wrote “these new Monosporascus spp. should be considered for a better formulation of integrated management strategies and breeding programs for soilborne fungi on cucurbit crops.”|
Is their actual report on the damages they cause in commercial fields? Way bothering developing and applied management strategies if they don’t cause any actual damages?
The keywords list can be updated.
Lines 47-49 – “M. eutypoides has a more limited geographical distribution, with only one unequivocal report in Tunisia from roots of watermelon and
cucumber crops (Cucumis sativus L.) [4].”
Is this may be the consequence of a lack of research instead of actual limited geographical distribution?
Lines 59-60 – “… temperature, hydrogen potential (pH), and salinity [1,3,11–22], which are key variables that influence the growth and development of disease‐associated fungal mycelia in plants.”
Is this true? What about oxidative stress generated by reactive oxygen species (ROS), superoxide anions, hydrogen peroxide, hydroxyl radicals, and nitrous oxide, which is part of the plant’s hypersensitive response (HR).
Line 70 – “Regarding salinity, M. cannonballus shows a high tolerance.” – you should note that different fungal structures (hyphae, spores, perithecia) may react differently to salinity stress (see for example Degani, O. and Goldblat Y., Ambient Stresses Regulate the Development of the Maize Late Wilt Causing Agent, Harpophora maydis. Agricultural Sciences (2014), 5 (7), 571-582.)
Line 153: What was the irrigation regime used?
Line 173: “a.i.” - Explain the term the first time it appears in the text - active ingredient.
Table 2: the regression formula and the R2 are not providing any essential information. Please replace these values with a graph and provide the R2 within the chart, for each plot.
Table 2: should contain only the species name, OpH, and EC50.
Table 3: describe “Rank” in the table footnotes.
Table 4: describe all abbreviations in the table footnotes. Specify the number of repeats and that the data are mean.
Table 5: as noted for Table 2, the regression formula and the R2 are not providing any essential information. Please replace these values with a graph and provide the R2 within the chart, for each plot.
Table 5: should contain only the species name and EC50 for each fungicide.
Lines 293-294 – “all cucurbits evaluated were susceptible to M. brasiliensis, M. caatinguensis, M. mossoroensis, M. nordestinus, and M. semiaridus”.
This is true only for the seedling stage under the pot experiment in a greenhouse! The result in the field may be different.
Author Response
Dear Reviewer 1,
We would like to thank you for the helpful comments and constructive observations. We have incorporated all the suggested revisions in the new version of the manuscript.
You commented about if there are reports of the damages that these new Monosporascus species cause in commercial fields. These new species have been only reported as weeds endophytes so far. We agree with you that this was too speculative. Thus, we have removed comments about integrated management strategies, but we have introduced information about recent reports of Monosporascus spp. as endophytes in USA and an associated research work that tried to answer questions about whether presumed endophytic lineages in the genus Monosporascus differ from pathogenic ones.
The Tables, Figures and the analyses of the results have been also modified according to your suggestions.
Best regards,
Dr. Rui Sales Júnior
Reviewer 2 Report
The purpose of this manuscript was to evaluate the response of new Monosporascus species to pH and salinity levels, to describe their sensitivity to several fungicides with different modes of action and to evaluate their pathogenicity to different cucurbits.
I am writing my review with mixed feeling. On one hand, the work was done properly bringing results. But on the other hand, I am asking myself what is the significance of those results ?
If a grower will find Monosporascus in his crop, he will not have the tools to know which Monosporascus is it, without checking it in the lab. The results of this article will be not applicable for him.
What is the new input that we learned from this work? We know for sure that in every pathogenic or non-pathogenic fungi we can found differences even in the same species. The differences are eco-types and not necessary new species. This is obvious that differences in pathogenicity, optimum temperature sensitivity to fungicides and many other characters will be slightly different in various isolates originated from different crops and different fields.
Again, the article is well done but I do not feel that it has significant contribution, thus I do not have any recommendation. The chief editor will have to decide whether to expect or decline this manuscript.
One technical question. In Table 2 and table 5. The authors gives values with 8 places after the point????? Why is that? Just makes it harder to look at the results.
- brasiliensis y = ‐0.01583333x3+0.306785714x2 ‐1.92166667x+5.027428571
Author Response
Dear Reviewer 2,
We would like to thank you for your comments and observations. We have prepared a new version of the manuscript following all reviewer’s comments.
We have incorporated some references mentioning recent research conducted in USA about Monosporascus endophytes in this country. This is a topic of interest because this information confirms that, in addition to cucurbit pathogens, the genus Monosporascus includes saprophytic species common in weeds or in plants growing in natural ecosystems in semiarid or arid regions. This arises questions about whether presumed endophytic lineages in the genus Monosporascus differ from pathogenic ones.
We consider that our work is a significant contribution to a better understanding and characterization of species diversity in this genus.
Best regards,
Dr. Rui Sales Júnior
Round 2
Reviewer 1 Report
Review report
Characterization of five new Monosporascus species: adaptation to environmental factors, pathogenicity to cucurbits and sensitivity to fungicides
The work includes a report on five recently described Monosporascus species, regarding mycelial growth at different pH levels and salinity concentrations, their pathogenicity to cucurbits sprouts (50 DAS), and their sensitivity to fungicides with different modes of action.
As I already stated in my first review, the manuscript consists of a lot of experimental work. The findings of this work are very interesting and important and contribute greatly to our knowledge about those Monosporascus species and their harmful potential as pathogens. The results advance our understanding of the pathogen isolates and their relationships with the host plant. The results are also essential to the development of disease management strategies.
General comments
The authors responded well to all the concerns raised in the first review.
The manuscript can be published in its current form, after a minor correction.
Figure 3 should be improved: in graph E, the Y label is incomplete – add (%). Also, write the letters (A-E) within each graph (at the upper left corner).
Author Response
REVIEWER 1
Dear Reviewer 1,
We would like to thank you again for the helpful comments and constructive observations. We have incorporated the minor revision in the new version of the manuscript.
Reviewer 1: Figure 3 should be improved: in graph E, the Y label is incomplete – add (%). Also, write the letters (A-E) within each graph (at the upper left corner).
Authors: OK
Best regards,
Dr. Rui Sales Júnior
Reviewer 2 Report
I appreciate the efforts made by the authors in correcting and improving the manuscript. Indeed, lot of good work was done but my feeling toward the manuscript was not changed.
The question I asking myself and I have to admit, I am not a taxonomist and do not know when a fungus is a new species or just a local eco-type, what is the significance of this study.
From the basic side, this manuscript provides data about differences between cucurbit pathogen different species but on the other hand, I am asking myself how such data provided in the lab and in pots experiments can help the growers.
I agree that this manuscript that was done properly, can be accepted for publication but it seems that its scientific value will be marginal.
Author Response
REVIEWER 2
Dear Reviewer 2,
We would like to thank you for accepting the manuscript, despite having doubts about the significance of this study.
We would like to confirm that the Monosporascus species included in this study are new species. Its description was supported by extensive morphological and molecular studies published at Annals of Applied biology journal (Negreiros, A.M.P.; Sales Júnior, R.; Rodrigues, A.P.M.S.; León, M.; Armengol, J. Prevalent weeds collected from cucurbit fields in Northeastern Brazil reveal new species diversity in the genus Monosporascus. Ann. Appl. Biol. 2019, 174, 349–363, doi:10.1111/aab.12493).
Best regards,
Dr. Rui Sales Júnior